# Effect of apolipoprotein genotype and educational attainment on cognitive function in autosomal dominant Alzheimer's disease

Stephanie Langella [1], N. Gil Barksdale[1], Daniel Vasquez [2], David Aguillon [2], Yinghua Chen[3], Yi Su [3], Natalia Acosta-Baena [2], Juliana Acosta-Uribe [2,4], Ana Y. Baena[2], Gloria Garcia-Ospina[2], Margarita Giraldo-Chica[2], Victoria Tirado[2], Claudia Muñoz[2], Silvia Ríos-Romenets[2], Claudia Guzman-Martínez[2], Gabriel Oliveira[1], Hyun-Sik Yang [5], Clara Vila-Castelar [1], Jeremy J. Pruzin[3], Valentina Ghisays [3], Joseph F. Arboleda-Velasquez [6], Kenneth S. Kosik [4], Eric M. Reiman [3,7], Francisco Lopera [2,7] & Yakeel T. Quiroz [1,2,7 ✉]

Autosomal dominant Alzheimer's disease (ADAD) is genetically determined, but variability in age of symptom onset suggests additional factors may influence cognitive trajectories. Although apolipoprotein E (*APOE*) genotype and educational attainment both influence dementia onset in sporadic AD, evidence for these effects in ADAD is limited. To investigate the effects of *APOE* and educational attainment on age-related cognitive trajectories in ADAD, we analyzed data from 675 Presenilin-1 E280A mutation carriers and 594 non-carriers. Here we show that age-related cognitive decline is accelerated in ADAD mutation carriers who also have an *APOE* e4 allele compared to those who do not and delayed in mutation carriers who also have an *APOE* e2 allele compared to those who do not. Educational attainment is protective and moderates the effect of *APOE* on cognition. Despite ADAD mutation carriers being genetically determined to develop dementia, age-related cognitive decline may be influenced by other genetic and environmental factors.

Presence of the e4 allele of the apolipoprotein E (*APOE*) gene is associated with increased risk for developing sporadic Alzheimer's disease (AD) and an earlier age of clinical onset than for individuals without an e4 allele[1,2]. However, evidence for an effect of *APOE* e4 genotype on cognitive function in autosomal dominant AD (ADAD) has been limited and inconclusive. ADAD is genetically determined by mutations on the amyloid precursor protein (*APP*), Presenilin-1 (*PSEN1*), and Presenilin-2

(*PSEN2*) genes[3]. The largest known kindred with ADAD due to a single mutation (*PSEN1* E280A) resides in Antioquia, Colombia. Carriers of this mutation have a median age of onset of mild cognitive impairment at 44 years and of dementia at 49 years[4]. Despite the group's well-characterized trajectory, there is individual variability in disease progression, highlighting the need to identify other genetic and environmental factors which may influence age-related cognitive decline.

[1]Massachusetts General Hospital, Harvard Medical School, Boston, MA, USA. [2]Grupo de Neurociencias de Antioquia, Facultad de Medicina, Universidad de Antioquia, Medellin, Colombia. [3]Banner Alzheimer's Institute, Phoenix, AZ, USA. [4]Neuroscience Research Institute and Department of Molecular, Cellular and Developmental Biology, University of California Santa Barbara, Santa Barbara, CA, USA. [5]Brigham and Women's Hospital, Harvard Medical School, Boston, MA, USA. [6]Schepens Eye Research Institute of Mass Eye and Ear, Harvard Medical School, Boston, MA, USA. [7]These authors jointly supervised this work: Eric M. Reiman, Francisco Lopera, Yakeel T. Quiroz. ✉e-mail: yquiroz@mgh.harvard.edu

The role of *APOE* e4 in this kindred has been inconclusive. In a previous study of 109 *PSEN1* E280A mutation carriers, those who had an *APOE* e4 allele had an earlier age of dementia onset than those who did not[5]. A subsequent study of 71 carriers in the same kindred found no effect of *APOE* e4, but found that the presence of the e2 allele was associated with delayed clinical onset by approximately eight years[6]. Broader investigations including ADAD carriers from multiple families have reported differences between *APP* and *PSEN1* mutations, which may mask *APOE* effects in combined analyses, but indicate detrimental effects on cognitive performance and decline in *PSEN1* carriers[7,8], demonstrating the importance of additional investigation in a large single kindred.

In addition, environmental factors (such as lifestyle, health, and socioeconomic conditions) may influence age-related cognitive trajectories and mitigate genetic risk[9–11]. One such factor is education (often defined as years of formal educational attainment), which has been identified as an important modifiable factor for dementia delay and prevention[12]. Higher educational attainment has been associated with slowed cognitive decline in older adults[13] and lower dementia incidence[14], indicating educational attainment promotes cognitive resilience in the face of pathology[15]. The reported impact of educational attainment in ADAD, however, is inconsistent. Lower educational attainment was a predictor of cognitive decline in ADAD due to various mutations[7] and of earlier clinical onset in *PSEN1* E280A carriers[16]. Unexpectedly, however, lower educational attainment (less than three years) has also been associated with later onset of dementia in carriers of the *PSEN1* E280A mutation[5]. Of note, ruralness was independently associated with both lower educational attainment and later age of onset in that sample[5], which may have contributed to the observed relationship. There have been no reported significant interactive effects between *APOE* genotype and educational attainment in ADAD to date.

Additional investigation in this kindred is required to clarify these discrepant findings. The role of genetic and environmental factors impacting age-related cognitive decline in ADAD are critical to further understand disease progression and to support future prevention and treatment goals. In this study, we aimed to evaluate the influence of *APOE* genotype on cognitive function in 675 *PSEN1* E280A carriers and 594 non-carrier family members, and secondarily explore whether educational attainment may be protective and moderate the relationship between *APOE* and cognitive function. We hypothesized that presence of the e4 allele would be associated with accelerated onset of cognitive impairment, presence of the e2 allele would be associated with delayed cognitive impairment, and that higher educational attainment would be protective against cognitive impairment.

Consistent with our hypotheses, in this work we show that the onset of age-related cognitive decline is accelerated in *PSEN1* E280A mutation carriers who are also *APOE* e4+ compared to those who are *APOE* e4− and delayed in those who are *APOE* e2+ compared to *APOE* e2−. Further, we find that educational attainment is protective and moderates the effect of *APOE* on cognition in *PSEN1* carriers.

## Results
### Sample characteristics
Sample characteristics are presented in Table 1. Of the 675 *PSEN1* E280A mutation carriers, 141 were *APOE* e4+ and 534 were *APOE* e4−. The *APOE* e4+ and e4− *PSEN1* E280A mutation carriers did not differ in age ($p = 0.64$) or sex ($p = 0.89$), and they did not differ in MMSE score ($p = 0.52$) when collapsing across age. *PSEN1* E280A mutation carriers who were also *APOE* e4+ had on average more years of educational attainment than those who were *APOE* e4− ($p = 0.02$). Of the 594 *PSEN1* E280A mutation non-carriers, 148 were *APOE* e4+ and 446 were *APOE* e4−. Within these non-carriers, *APOE* e4+ and e4− individuals did not differ in age ($p = 0.42$), sex ($p = 0.23$), educational attainment ($p = 0.42$), or MMSE score ($p = 0.82$).

### Age-related cognitive function by *PSEN1* and *APOE* e4 genotype
We first estimated the age-related trajectory of cognitive impairment, measured through MMSE total score, using the Hamiltonian Markov chain Monte Carlo method in *PSEN1* E280A mutation carriers and non-carriers, irrespective of APOE genotype. MMSE was negatively associated with age in *PSEN1* carriers and significantly differentiated carriers from non-carriers at 31.5 years (Fig. 1).

We then estimated the age-related trajectory of cognitive impairment as a function of *APOE* e4 genotype separately in *PSEN1* E280A mutation carriers and non-carriers. The cognitive trajectories of *APOE* e4+ and e4− *PSEN1* E280A mutation carriers diverged at 44.3 years, approximately the median age of onset of mild cognitive impairment in this kindred[4] (Fig. 2A, B). In contrast, the age-related cognitive trajectories of *APOE* e4+ and e4− *PSEN1* E280A mutation non-carriers did not diverge (Fig. 2C, D). To supplement these analyses, age of clinical onset was compared in a subset of *PSEN1* mutation carriers who had converted to MCI or dementia. Consistent with the prior findings, *PSEN1* mutation carriers who were also *APOE* e4+ had earlier ages of clinical onset compared to those who were *APOE* e4− (Supplementary Table 1).

### Role of educational attainment on cognition in *PSEN1* E280A carriers
Years of educational attainment was examined as a protective and potentially modifying factor of the relationship between *APOE* e4 and cognitive function using separate linear regressions for *PSEN1* E280A mutation carriers and non-carriers. Within *PSEN1* E280A mutation carriers, being *APOE* e4+ was associated with lower MMSE scores compared to *APOE* e4− ($\beta = -3.37$, $p = 0.001$). Irrespective of *APOE* genotype, higher educational attainment was associated with higher MMSE scores ($\beta = 0.41$, $p < 0.001$). There was also a significant interaction between *APOE* e4 and years of educational attainment ($\beta = 0.32$, $p = 0.005$; Fig. 3a) such that the negative effect of *APOE* e4+ was attenuated as years of educational attainment increased. In other words, higher levels of educational attainment mitigated the additional risk conferred by the presence of at least one e4 allele in *PSEN1* E280A carriers.

In non-carriers of the *PSEN1* E280A mutation, there was no significant effect of *APOE* e4 on MMSE score ($\beta = -0.17$, $p = 0.66$), but there was a significant main effect of educational attainment ($\beta = 0.16$, $p < 0.001$) such that higher educational attainment was associated with higher MMSE scores. There was no interaction between *APOE* e4 and educational attainment ($\beta = -0.003$, $p = 0.94$).

### Age-related cognitive function by *PSEN1* and *APOE* e2 genotype
Our findings suggest that within *PSEN1* E280A carriers, age-related cognitive decline begins earlier in those who are *APOE* e4+ than for those with other *APOE* genotypes, including those who are homozygous e3 and e2+. Because presence of the e2 allele has been associated with delayed clinical onset in this kindred[6], we sought to examine the association between *APOE* e2 and cognition in our current sample. Of the 675 *PSEN1* E280A mutation carriers, 102 were *APOE* e2+ and 573 were *APOE* e2− (Supplementary Table 2). The *APOE* e2+ and e2− *PSEN1* E280A mutation carriers did not differ in age ($p = 0.20$), sex ($p = 0.65$), educational attainment ($p = 0.69$), or MMSE score ($p = 0.37$). Of the 594 *PSEN1* E280A mutation non-carriers, 73 were *APOE* e2+ and 521 were *APOE* e2−, and *APOE* e2+ and e2− individuals did not differ in age ($p = 0.47$), sex ($p = 0.12$), educational attainment ($p = 0.70$), or MMSE score ($p = 0.66$).

We first estimated the age-related trajectory of cognitive impairment as a function of *APOE* e2 genotype separately in *PSEN1* E280A mutation carriers and non-carriers (Supplementary Fig. 1a, b). The cognitive trajectories of *APOE* e2+ and e2− *PSEN1* E280A mutation carriers diverged at 41.1 years, such that *APOE* e2+ *PSEN1* carriers had delayed age-related global cognitive decline. Age of clinical onset of

**Table 1 | Participant demographics stratified by *PSEN1* and *APOE* genotype**

| | *PSEN1* E280A carriers | | | *PSEN1* E280A non-carriers | | |
|---|---|---|---|---|---|---|
| | *APOE* e4+ | *APOE* e4− | p | *APOE* e4+ | *APOE* e4− | p |
| N | 141 | 534 | | 148 | 446 | |
| Age | 33.70 ± 10.89 | 34.25 ± 11.27 | 0.64 | 35.53 ± 11.78 | 34.83 ± 12.07 | 0.42 |
| Sex (M/F) | 63/78 | 242/292 | 0.89 | 59/89 | 203/243 | 0.23 |
| Educational attainment (years) | 8.10 ± 3.96 | 7.25 ± 4.47 | 0.02 | 8.44 ± 4.76 | 8.17 ± 4.68 | 0.42 |
| Mini Mental State Examination | 26.16 ± 5.56 | 26.56 ± 5.31 | 0.52 | 28.64 ± 2.25 | 28.78 ± 2.05 | 0.82 |

Means and standard deviations given for age, educational attainment, and Mini Mental State Examination; uncorrected p values from two-sided statistical comparisons (Mann–Whitney U tests used to compare age, educational attainment, and Mini Mental State Examination scores; chi-square test used to compare sex).

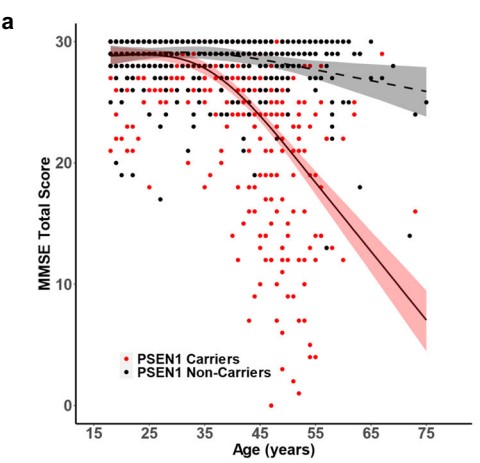
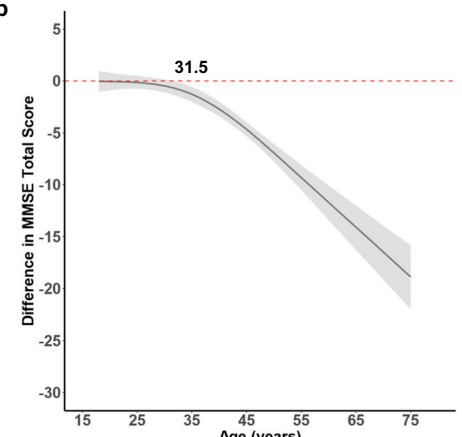

**Fig. 1 | Age-related trajectories of cognitive impairment in *PSEN1* E280A mutation carriers and non-carriers. a** Cross-sectional MMSE scores of *PSEN1* E280A mutation carriers (red) and non-carriers (black) as a function of age. **b** Differences in MMSE score between *PSEN1* E280A mutation carriers and non-carriers as a function of age. MMSE score declines in mutation carriers begins to differ from non-carriers at 31.5 years. The shaded areas of each plot represent the 99% credible intervals around the model estimates drawn from the distributions of model fits derived by the Hamiltonian Markov chain Monte Carlo analyses. MMSE Mini Mental State Examination. Source data are provided as a Source Data file.

the MCI and dementia converters did not significantly differ between *APOE* e2+ and e2− genotypes, but the trends were in the hypothesized direction such that *APOE* e2+ mutation carriers had on average later clinical onset (Supplementary Table 1). In *PSEN1* E280A mutation non-carriers, the age-related cognitive trajectories of *APOE* e2+ and e2− individuals did not diverge (Supplementary Fig. 1c, d).

We then assessed the relationships between *APOE* e2 genotype and educational attainment on global cognitive function. Within *PSEN1* E280A mutation carriers, there was a main effect of genotype, such that those who were *APOE* e2− had lower MMSE scores than those who were *APOE* e2+ (ß = −2.78, p = 0.007). Across *APOE* genotype, higher educational attainment was associated with higher MMSE scores (ß = 0.26, p = 0.018). Additionally, there was a significant interaction between *APOE* e2 genotype and educational attainment, ß = 0.24, p = 0.046 (Fig. 3b), such that higher educational attainment attenuated the negative effect of being *APOE* e2−.

## Discussion

*APOE* e4 has long been associated with increased risk and earlier age of onset for sporadic AD[1,2,17–20]. A rare variant on the *APOE* e3 allele was found to delay onset of MCI by three decades in a *PSEN1* E280A carrier[21], yet the evidence linking the more common e2 and e4 variants has been mixed[5,6,22]. Our findings in over 1,000 participants from a single kindred show an added effect of *APOE* genotype in carriers of the *PSEN1* E280A mutation for ADAD, such that *APOE* e4+ *PSEN1* mutation carriers had accelerated onset of age-related cognitive decline compared to *APOE* e4− *PSEN1* mutation carriers. The age-related trajectory of clinical impairment diverged between *APOE* e4+

and e4− *PSEN1* mutation carriers around age 44, approximately the median age of onset of mild cognitive impairment in this kindred[4]. Our results are consistent with a prior study reporting a detrimental effect of *APOE* e4+ genotype in *PSEN1* E280A mutation carriers[5]. A subsequent study of 71 *PSEN1* E280A mutation carriers found no effect of *APOE* e4 genotype, but the relationship was in the expected negative direction[6], suggesting the study may have been underpowered to detect an effect. Conversely, we found that *PSEN1* E280A mutation carriers who were *APOE* e2+ had delayed onset of age-related cognitive decline, replicating a prior finding from this kindred[6]. The *APOE* e2 allele has also been associated with delayed onset and protection against cognitive decline in older adults[20,23,24].

More research is needed to determine how these genetic risk factors contribute to earlier cognitive decline in ADAD. Both *APOE* e4 and *PSEN1* mutations influence accumulation of β-amyloid (Aβ) pathology in the brain[25–27]. This genetic combination may result in earlier or higher pathological burden, but future studies will need to consider age-related trajectories of brain pathology in addition to clinical impairment. Since other mutations causing ADAD also alter Aβ production, similar results might be expected in other ADAD mutations. However, the extent to which these results generalize to ADAD caused by other mutations is uncertain. A study of *APP* and *PSEN1* mutation carriers from six families found no overall effect of *APOE* e4 on cognitive decline, but in a direct comparison of age-related cognitive decline of *APOE* e4+ *APP* and *PSEN1* carriers, the *APOE* e4+ *PSEN1* carriers had a steeper decline than the *APOE* e4+ *APP* carriers[7].

Additional questions remain about the nuances of *APOE* genotype in ADAD. It is currently unknown whether the risk is greater in

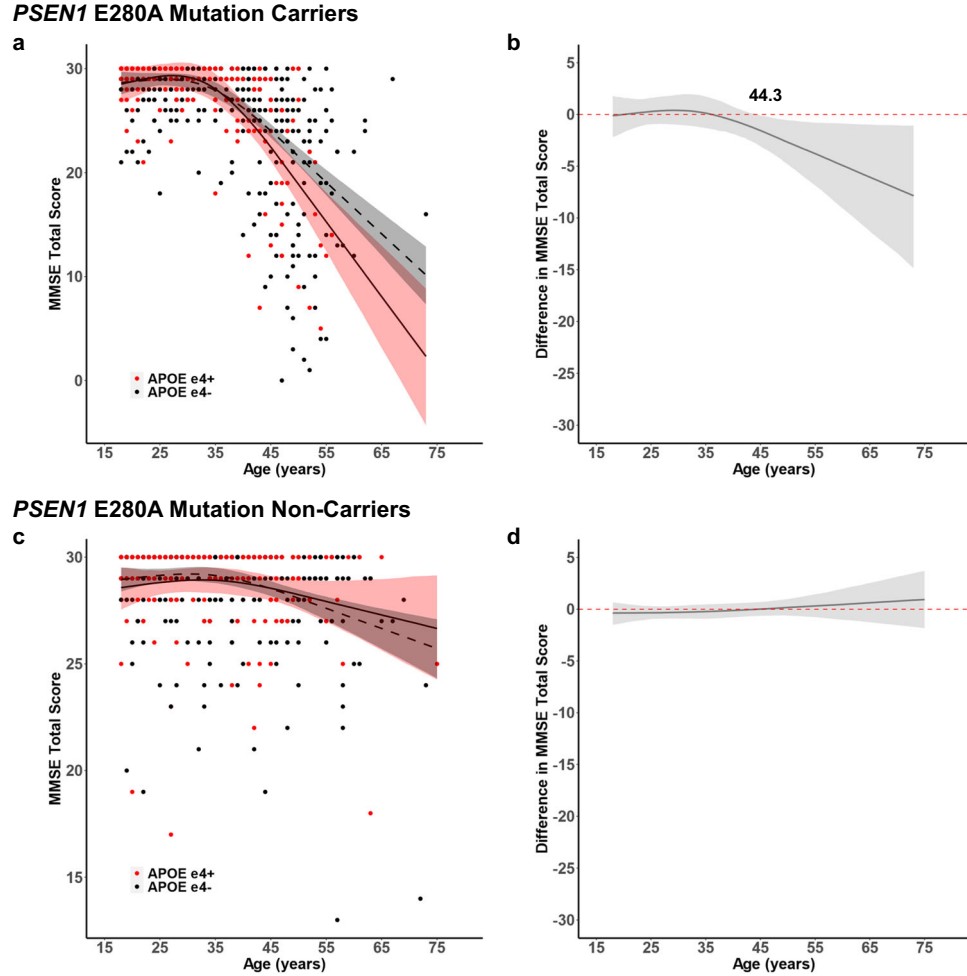

**Fig. 2 | Age-related trajectories of cognitive impairment in *PSEN1* E280A mutation carriers and non-carriers stratified by presence or absence of *APOE* e4. a** Cross-sectional MMSE scores of *PSEN1* E280A mutation carriers who are *APOE* e4+ (red) and *APOE* e4- (black) as a function of age. **b** Differences in MMSE score between *APOE* e4+ and e4− *PSEN1* E280A mutation carriers as a function of age. MMSE score declines in *APOE* e4+ *PSEN1* E280A mutation carriers begins to differ from *APOE* e4− *PSEN1* E280A mutation carriers at 44.3 years. **c** Cross-sectional MMSE scores of *PSEN1* E280A mutation non-carriers who are *APOE* e4+ (red) and

*APOE* e4− (black) as a function of age. **d** Differences in MMSE score between *APOE* e4+ and e4- *PSEN1* E280A mutation non-carriers as a function of age. MMSE score does not differ between *APOE* e4+ and e4− *PSEN1* E280A mutation non-carriers in this age range. The shaded areas of each plot represent the 99% credible intervals around the model estimates drawn from the distributions of model fits derived by the Hamiltonian Markov chain Monte Carlo analyses. MMSE = Mini Mental State Examination. Source data are provided as a Source Data file.

homozygous *APOE* e4 ADAD mutation carriers than in heterozygous *APOE* e4 ADAD mutation carriers, as is observed in sporadic AD[2]. This question is particularly challenging to answer in ADAD given the small percentage of homozygous *APOE* e4 carriers coupled with the small population of ADAD mutation carriers. Observational evidence from a sample of 17 carriers of an *APP* mutation supports this notion, such that homozygous *APOE* e4 mutation carriers had the earliest age of onset, followed by the heterozygous *APOE* e4 mutation carriers, and finally the *APOE* e2 mutation carriers with the latest onset[28]. In a comparison of early- and late-onset AD, *APOE* e4 genotype was associated with accelerated cognitive decline in both groups[29]. Coupled with the current results, these findings provide converging evidence that *APOE* may have similar effects in sporadic and autosomal dominant AD.

Furthermore, in sporadic AD, *APOE* e4 has been associated with better cognitive performance and differing neural activity during young adulthood[30–32], reflecting possible antagonistic pleiotropy of this genetic risk factor[33,34]. Structurally, *APOE* e4+ adults show greater parahippocampal thickness than *APOE* e4− adults[35] and differences in white matter integrity that may relate to observed cognitive benefits[36]. Our study only considered age-related cognitive trajectories in adult carriers; however, it is possible that *APOE* e4 may provide some

biological or cognitive benefit in younger *PSEN1* carriers (i.e., childhood, akin to young adults in sporadic AD). In fact, *PSEN1* carriers in this kindred have greater cortical thickness in childhood than non-carriers, followed by atrophy in adulthood[37]. Studying the influence of both *APOE* and *PSEN1* across the lifespan will enhance our knowledge of the effects of these genes, and, hopefully, provide mechanisms for disease prevention and treatment in the future.

Despite the additional risk conferred by *APOE* e4, our results suggest that educational attainment may be a critical mechanism of cognitive reserve in ADAD, as previously shown in sporadic AD[13–15]. Higher educational attainment was related to higher global cognition in *PSEN1* carriers and mitigated the cognitive impairment associated with *APOE* e4. Prior studies in this kindred have found opposing results, including one in which more years of educational attainment was associated with delayed clinical onset[16], and one in which higher educational attainment (defined categorically as greater than three years) was associated with lower cognition[5] in *PSEN1* mutation carriers. The conflicting results may arise from differences in the samples' average years of educational attainment, the treatment of educational attainment as a continuous versus categorical variable, or from confounding variables influencing these relationships. In the prior study

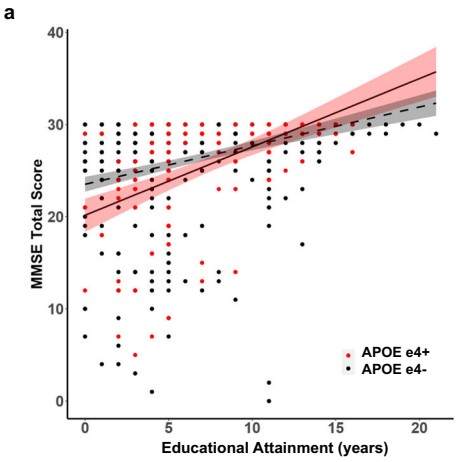
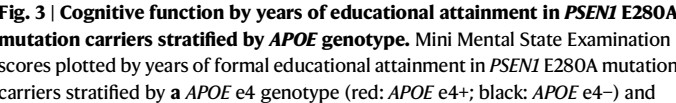
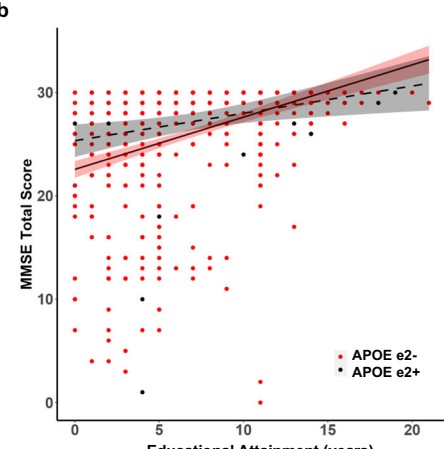

**Fig. 3 | Cognitive function by years of educational attainment in *PSEN1* E280A mutation carriers stratified by *APOE* genotype.** Mini Mental State Examination scores plotted by years of formal educational attainment in *PSEN1* E280A mutation carriers stratified by **a** *APOE* e4 genotype (red: *APOE* e4+; black: *APOE* e4−) and **b** *APOE* e2 genotype (red: *APOE* e2−; black: *APOE* e2+). Plots show regression line with shaded standard error bands. MMSE = Mini Mental State Examination. Source data are provided as a Source Data file.

reporting a detrimental effect of educational attainment, low educational attainment was highly correlated with ruralness, which similarly was associated with later clinical onset[5]. Ruralness, then, may reflect protective factors (e.g., physical activity, environment) that explain the reported positive association of low educational attainment. We similarly found that ruralness was associated with fewer years of educational attainment in our sample, but we did not find an association between ruralness and global cognition (see Supplementary Analysis). The *PSEN1* carriers and non-carriers of our sample are members of the same families, providing a high degree of environmental matching, although additional variables explaining quality rather than quantity of years of educational attainment may contribute to our findings and will be important to consider in future work.

Many studies examining education effects in sporadic AD include older adult populations with high levels of educational attainment, with averages often greater than high school or college. This is not representative of the broader population, and there are many conflicting findings in the role of educational attainment on cognitive reserve in older adults[13,14,38,39]. In contrast, our sample included a broad range of educational attainment. Our results indicate that low levels of formal educational attainment, in particular, confers greater risk. As such, programs to increase early years of education may be particularly important as preventative measures, supported by the inclusion of education as one of the 12 modifiable risk factors for dementia in the most recent Lancet commission[12]. The factors contributing to higher versus lower educational attainment (e.g., socioeconomic status, occupational attainment) as well as the underlying biological mechanisms of this *APOE*-educational attainment interaction require further study, particularly since higher levels of educational attainment have been associated with lower Aβ in ADAD[40].

A primary limitation of this study is the reliance of single time-point data. Although our participants spanned a broad range of ages, and age is highly linked to clinical progression in this kindred, we cannot speak to individual trajectories of decline with these data. Analysis of longitudinal cognitive decline will further clarify whether *APOE* influences age of onset, rate of decline, or both. Additionally, despite having one of the largest sample sizes compared to prior literature in ADAD, our study was underpowered to assess potential gene dose-dependent effects of the *APOE* e4 or e2 alleles (see Supplementary Fig. 2 for visualization of age-related cognitive trajectories). Finally, educational attainment is not the sole environmental factor influencing cognition and clinical progression. More work is needed to

further understand the impact of other lifestyle and modifiable factors along with their interactions with genetic makeup.

Together, our results highlight the importance of studying additional genetic and environmental risk factors in ADAD populations, with critical implications for future disease prevention and interventions. Future studies can push these questions forward by investigating the biological basis for the additive risk of *APOE* e4 and *PSEN1* mutations and for the protective role of *APOE* e2, and for the aspects and length of educational attainment that can support cognitive function or reduce the risk of dementia. Inclusion of blood-based biomarkers in such studies characterizing disease progression is necessary to increase access in these populations at risk and to understand the biological mechanisms underlying these findings[41–43]. The answers to these questions will inform how to best implement educational interventions in various communities and whether continuing late life education may provide additional protection. These answers are critical as Aβ- and *APOE*-based treatments for AD are being investigated[44,45].

In conclusion, our results demonstrate that (1) age-related changes in global cognitive function may be accelerated in ADAD mutation carriers who are also *APOE* e4+ compared to those who are *APOE* e4−; (2) age-related changes in global cognitive function may be delayed in ADAD mutation carriers who are also *APOE* e2+ compared to those who are *APOE* e2−; and (3) higher educational attainment may have a protective effect against cognitive impairment, even in the presence of strong genetic risk factors.

## Methods

Study procedures were approved by the Institutional Review Board of the University of Antioquia in Colombia (21-10-605) and were performed in accordance with the ethical standards of the Declaration of Helsinki. All participants provided informed consent prior to the initiation of study procedures. Participants were compensated for their participation in accordance with the approved guidelines.

### Study design and participants

This cross-sectional study included participants over the age of 18 recruited from the Alzheimer's Prevention Initiative (API) registry of ADAD, which includes more than 6000 living members of a kindred with a high prevalence of carriers of the *PSEN1* E280A mutation (approximately 1200 individuals)[46]. All members of the registry reside in Colombia and have a parent with the *PSEN1* E280A mutation but are

blind to their own genetic status. In total, 675 *PSEN1* E280A mutation carriers (370 female, 305 male) and 594 mutation non-carriers (332 female, 262 male) were included in these analyses.

Neuropsychological assessments were performed at the University of Antioquia in Colombia. Participants completed a clinical interview and the Mini Mental State Examination (MMSE), administered in Spanish, used as a proxy for cognitive impairment. Cognitive data were stored using REDCap (v. 13.1.29). Investigators were blind to participant genetic status during data collection.

## Genotyping

Genomic DNA was extracted from the blood by standard protocols, and *PSEN1* E280A characterization was done at the University of Antioquia using methods previously described[47]. Genomic DNA was amplified with the primers *PSEN1*-S 5′ AACAGCTCAGGAGAGGAATG 3′ and PSEN1-AS 5′ GATGAGACAAGTNCCNTGAA 3′. We used the restriction enzyme *Bsm*I for restriction fragment length polymorphism analysis. Each participant was classified as a *PSEN1* E280A carrier or non-carrier.

*APOE* genotyping was performed using a Kompetitive Allele Specific PCR–KASP™ assay[48] (LGV Genomics, Beverly, MA). Due to low numbers of homozygous e4 carriers and homozygous e2 carriers (see Table 2), each participant was classified based on the presence of at least one e4 allele (e4+) or no e4 alleles (e4−), and separately, based on the presence of at least one e2 allele (e2+) or no e2 alleles (e2−). Twenty-nine participants (13 *PSEN1* carriers and 16 non-carriers) were *APOE* e2/e4 and, therefore, included in both *APOE* e4+ and *APOE* e2+ groups.

## Statistical analysis

All analyses were conducted in R version 4.2.0, modeled separately for *PSEN1* E280A carriers and non-carriers. Group differences in continuous variables were assessed using Mann–Whitney *U* tests due to non-normality, and dichotomous variables were compared using chi-square tests. *APOE* genotype was included in analyses as a dichotomous variable. Cognitive impairment was measured through MMSE total score (maximum score = 30). Age-related trajectories were derived from cross-sectional MMSE scores modeled using a restricted cubic spline model. Model parameters were estimated using a Hamiltonian Markov chain Monte Carlo method to compare group trajectories (prior = Cauchy distribution, chains = 8, iterations = 10,000, thin = 10). Linear regression was used to estimate the effect of educational attainment on cognition, with MMSE total score as the dependent variable and *APOE* genotype, educational attainment, and their interaction term as predictors. Educational attainment was included as a continuous variable, representing self-reported total years of formal educational attainment. Self-reported sex was collected for each participant and is presented in demographic tables. Sex was not included in statistical analyses due to no a priori hypotheses about sex differences and sample size limitations of subdividing the participants by *PSEN1* genotype, *APOE* genotype, and sex; however, the proportions of

males and females were roughly similar, and there were no differences in sex distributions in the comparison groups of interest, thus we believe results are generalizable to both males and females.

## Reporting summary

Further information on research design is available in the Nature Portfolio Reporting Summary linked to this article.

## Data availability

Anonymized clinical, cognitive and genetic data are available upon request, subject to an internal review by F.L., and Y.T.Q. to ensure that participant confidentiality and *PSEN1* E280A carrier or non-carrier status are protected, completion of a data sharing agreement and in accordance with the University of Antioquia's and MGH's institutional review board and institutional guidelines. Please submit requests for participant-related data to Y.T.Q. (yquiroz@mgh.harvard.edu). Source data are provided with this paper.

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

**Table 2 | Distribution of *APOE* genotypes for *PSEN1* E280A carriers and non-carriers**

|  | *PSEN1* E280A Carriers (n = 675) | *PSEN1* E280A Non-Carriers (n = 594) |
|---|---|---|
| e2/e2 | 4 (0.59%) | 2 (0.34%) |
| e2/e3 | 85 (12.59%) | 55 (9.26%) |
| e2/e4 | 13 (1.93%) | 16 (2.69%) |
| e3/e3 | 445 (65.93%) | 389 (65.49%) |
| e3/e4 | 117 (17.33%) | 122 (20.54%) |
| e4/e4 | 11 (1.63%) | 10 (1.68%) |

17. William Rebeck, G., Reiter, J. S., Strickland, D. K. & Hyman, B. T. Apolipoprotein E in sporadic Alzheimer's disease: allelic variation and receptor interactions. *Neuron* **11**, 575–580 (1993).

18. Poirier, J. et al. Apolipoprotein E polymorphism and Alzheimer's disease. *Lancet* **342**, 697–699 (1993).

19. Ashford, J. W. APOE genotype effects on Alzheimer's disease onset and epidemiology. *J. Mol. Neurosci.* **23**, 157–166 (2004).

20. Reiman, E. M. et al. Exceptionally low likelihood of Alzheimer's dementia in APOE2 homozygotes from a 5,000-person neuro-pathological study. *Nat. Commun.* **11**, 667 (2020).

21. Arboleda-Velasquez, J. F. et al. Resistance to autosomal dominant Alzheimer's disease in an APOE3 Christchurch homozygote: a case report. *Nat. Med.* **25**, 1680–1683 (2019).

22. Van Broeckhoven, C. et al. APOE genotype does not modulate age of onset in families with chromosome 14 encoded Alzheimer's disease. *Neurosci. Lett.* **169**, 179–80 (1994).

23. Suri, S., Heise, V., Trachtenberg, A. J. & Mackay, C. E. The forgotten APOE allele: a review of the evidence and suggested mechanisms for the protective effect of APOE ε2. *Neurosci. Biobehav. Rev.* **37**, 2878–2886 (2013).

24. Corder, E. H. et al. Protective effect of apolipoprotein E type 2 allele for late onset Alzheimer disease. *Nat. Genet.* **7**, 180–184 (1994).

25. Lemere, C. A. et al. The E280A presenilin 1 Alzheimer mutation produces increased Aβ42 deposition and severe cerebellar pathology. *Nat. Med.* **2**, 1146–1150 (1996).

26. Wildsmith, K. R., Holley, M., Savage, J. C., Skerrett, R. & Landreth, G. E. Evidence for impaired amyloid β clearance in Alzheimer's disease. *Alzheimer's Res. Ther.* **5**, 33 (2013).

27. Ye, S. et al. Apolipoprotein (apo) E4 enhances amyloid β peptide production in cultured neuronal cells: ApoE structure as a potential therapeutic target. *Proc. Natl Acad. Sci. USA* **102**, 18700–18705 (2005).

28. Sorbi, S. et al. Epistatic effect of APP717 mutation and apolipoprotein E genotype in familial Alzheimer's disease. *Ann. Neurol.* **38**, 124–127 (1995).

29. Polsinelli, A. J. et al. APOE ε4 carrier status and sex differentiate rates of cognitive decline in early- and late-onset Alzheimer's disease. *Alzheimer's Dement.* https://doi.org/10.1002/alz.12831 (2022).

30. Evans, S. et al. Cognitive and neural signatures of the APOE E4 allele in mid-aged adults. *Neurobiol. Aging* **35**, 1615–1623 (2014).

31. Rusted, J. M. et al. APOE e4 polymorphism in young adults is associated with improved attention and indexed by distinct neural signatures. *Neuroimage* **65**, 364–373 (2013).

32. Dennis, N. A. et al. Temporal lobe functional activity and connectivity in young adult APOE ε4 carriers. *Alzheimer's Dement.* **6**, 303–311 (2010).

33. Tuminello, E. R. & Han, S. D. The apolipoprotein e antagonistic pleiotropy hypothesis: review and recommendations. *Int. J. Alzheimers Dis.* **2011**, 726197 (2011).

34. Gharbi-Meliani, A. et al. The association of APOE ε4 with cognitive function over the adult life course and incidence of dementia: 20 years follow-up of the Whitehall II study. *Alzheimers Res. Ther.* **13**, 5 (2021).

35. Dowell, N. G. et al. Structural and resting-state MRI detects regional brain differences in young and mid-age healthy APOE-e4 carriers compared with non-APOE-e4 carriers. *NMR Biomed.* **29**, 614–624 (2016).

36. Dowell, N. G. et al. MRI of carriers of the apolipoprotein E e4 allele-evidence for structural differences in normal-appearing brain tissue in e4+ relative to e4- young adults. *NMR Biomed.* **26**, 674–82 (2013).

37. Fox-Fuller, J. T. et al. Cortical thickness across the lifespan in a Colombian cohort with autosomal-dominant Alzheimer's disease: a cross-sectional study. *Alzheimer's Dement.* **13**, e12233 (2021).

38. Wilson, R. S. et al. Education and cognitive reserve in old age. *Neurology* **92**, e1041–e1050 (2019).

39. Sharp, E. S. & Gatz, M. Relationship between education and dementia. *Alzheimer Dis. Assoc. Disord.* **25**, 289–304 (2011).

40. Gonneaud, J. et al. Association of education with Aβ burden in preclinical familial and sporadic Alzheimer disease. *Neurology* **95**, e1554–e1564 (2020).

41. Leuzy, A., Cullen, N. C., Mattsson-Carlgren, N. & Hansson, O. Current advances in plasma and cerebrospinal fluid biomarkers in Alzheimer's disease. *Curr. Opin. Neurol.* **34**, 266–274 (2021).

42. Telser, J., Risch, L., Saely, C. H., Grossmann, K. & Werner, P. P-tau217 in Alzheimer's disease. *Clin. Chim. Acta* **531**, 100–111 (2022).

43. Aguillon, D. et al. Plasma p-tau217 predicts in vivo brain pathology and cognition in autosomal dominant Alzheimer's disease. *Alzheimers Dement.* https://doi.org/10.1002/alz.12906 (2022).

44. Wisniewski, T. & Drummond, E. APOE-amyloid interaction: therapeutic targets. *Neurobiol. Dis.* **138**, 104784 (2020).

45. Williams, T., Borchelt, D. R. & Chakrabarty, P. Therapeutic approaches targeting Apolipoprotein E function in Alzheimer's disease. *Mol. Neurodegener.* **15**, 8 (2020).

46. Reiman, E. M. et al. Alzheimer's prevention initiative: a plan to accelerate the evaluation of presymptomatic treatments. *J. Alzheimer's Dis.* **26**, 321–329 (2011).

47. Lendon, C. L. et al. E280A PS-1 mutation causes Alzheimer's disease but age of onset is not modified by ApoE alleles. *Hum. Mutat.* **10**, 186–195 (1997).

48. He, C., Holme, J. & Anthony, J. SNP genotyping: the KASP Assay BT - crop breeding: methods and protocols. in *Methods in Molecular Biology* (eds. Fleury, D. & Whitford, R.) 75–86 (Springer New York, 2014).

## Acknowledgements

The authors thank the *PSEN1* Colombian families for contributing their valuable time and effort, without which this study would not have been possible. We thank the research staff of the Group of Neuroscience of Antioquia for their help coordinating study visits for the Colombian API Registry. We thank Geidy Serrano from the Banner institute for her help quantifying DNA samples. This study was supported by grant DP5OD019833 from the National Institutes of Health Office of the Director (Y.T.Q.), the Massachusetts General Hospital Executive Committee on Research (Y.T.Q.), and grant R01AG054671 from the National Institute on Aging (Y.T.Q.). The funders had no role in the design and conduct of the study; collection, management, analysis, and interpretation of the data; preparation, review, or approval of the manuscript; and decision to submit the manuscript for publication.

## Author contributions

E.M.R., F.L., and Y.T.Q. initiated this work, directed and supervised conduction of the study. S.L, N.G.B and Y.T.Q. drafted the manuscript. Clinical information was collected and analyzed by F.L., D.V., D.A., N.A-B, A.B., M.G-C, V.T., C.M., S.R-R., and C.V.-C. Genetic data was collected and analyzed by G.G., C.G-M., J.A.-U., and K.K. Statistical analyses were conducted by Y.C., V.G, and Y.S. All authors revised and contributed to finalize the manuscript.

## Competing interests

S.L. is supported by a grant from the Alzheimer's Association (AARF-22-920754). Y.S. reports grants from The Alzheimer's Association, The BrightFocus Foundation, NIH/NIA, State of Arizona, outside the submitted work. C.V.-C. reports grants from the Alzheimer's Association (AARF 2019A005859) and the National Institute on Aging (K99AG073452). K.S.K. is on the Board of Directors for the Tau Consortium, receives funding from the NIA, the Alzheimer Association, and the Alzheimer's Drug Discovery Foundation. H-S.Y. reports a grant from the National Institute of Aging (K23 AG062750). E.M.R. reports grants from National Institute on Aging (P30 AG072980, R01 AG069453, R01 AG055444), Banner Alzheimer's Foundation and the NOMIS Foundation

during the conduct of the study. E.M.R. is a compensated scientific advisor for Alzheon, Aural Analytics, Denali, Retromer Therapeutics, and Vaxxinity, an uncompensated scientific advisor for Lilly, and a cofounder, advisor and shareholder of AlzPATH, which is involved in the development of blood-based biomarkers for Alzheimer's disease outside the scope of the submitted. In addition, E.M.R. is the inventor of a patent issued to Banner Health, which involves the use of biomarker endpoints in at-risk persons to accelerate the evaluation of Alzheimer's disease prevention therapies and is outside the submitted work. F.L. was supported by an Anonymous Foundation, and the Administrative Department of Science, Technology and Innovation (Colciencias Colombia;111565741185). E.M.R. and F.L. are principal investigators of the Alzheimer's Prevention Initiative (API) Autosomal Dominant AD Trial, which is supported by NIA, philanthropy, Genentech, and Roche. Y.T.Q. was supported by grants from the National Institute on Aging (R01 AG054671, RF1AG077627), the Alzheimer's Association, and Massachusetts General Hospital ECOR. Y.T.Q. serves as consultant for Biogen. The remaining authors declare no competing interests.
