## [Peer review file · Nature Communications]

REVIEWER COMMENTS

Reviewer #1 (Remarks to the Author):

It is interesting to know the effects of apoe on age at onset and rate of decline in PSEN mutation carriers and this has been addressed before with conflicting results (see also Van Broeckhoven C, Backhovens H, Cruts M, Martin JJ, Crook R, Houlden H, Hardy J. APOE genotype does not modulate age of onset in families with chromosome 14 encoded Alzheimer's disease. *Neurosci Lett.* 1994 Mar 14;169(1-2):179-80. doi: 10.1016/0304-3940(94)90385-9. PMID: 8047278.). This is the largest study to date.

I am not sure however, the authors have correctly interpreted their data. Does apoe genotype influence age at onset (which I would suspect) or rate of decline (which would surprise me). Most studies in non mutation carriers suggest that apoe affects a.o.o. but not rate of decline: ie.disease initiation.

I would suggest reanalysis to distinguish these possibilities

Reviewer #2 (Remarks to the Author):

The study investigated the effect of APOE e4 and e2 status as well as years of education on onset of dementia and cognitive function (MMSE) in a huge cohort of mutation carriers (MC) and non-carriers (NC) from the Colombian PSEN1 E280A kindred. In MC, the results showed a negative e4 and a positive e2 effect and a positive influence of education and the effects were age-related after the onset. In contrast, no significant effect were found in NC.

Given the cross-sectional design, results clarifies previous inconsistencies regarding the e4 effect in the Colombian kindred.

In spite of the robustness of the study, some questions about the cohort have come to mind. What was the age range and what was the age distribution? Could the total cohort be divided in two to make it possible study cohort effects? Could it be that age changes are confounded by cohort changes regarding education, way of living etc? This point was raised by the authors in the discussion.

A related issue regards "ruralness" that was also mentioned in the discussion as possible important factor to understand differential effects of e4 (positive vs negative). It would be informative to know

about urban vs rural living conditions in the cohort. This factor may also be confounded by degree of education.

In regard to a possible differential effect of e4 on young and old individuals, the concept of "antagonistic pleiotropy" was referred to without presenting a reference. It maybe important for understanding the variability of APOE effects.

Several times (introduction, discussion), the authors use the word "environmental", what is meant by this umbrella term? Pollution, virus, social conditions (SES)? It seems obvious that other factors than AD and APOE genetics and education are players as unknown influence is operating, see Figures 1A, 2A, 3A and 3B (all the dots below regression lines).

Reviewer #3 (Remarks to the Author):

APOE4 and APOE2 genotype have been identified as a major genetic risk and protective factor, respectively, for late-onset sporadic Alzheimer's disease (AD). However, their effects on autosomal dominant AD (ADAD) are unclear. Previous studies of APOE genotype effects on ADAD yielded inconsistent data due to the under-power analysis related to small sample sizes.

To address this major issue, the authors reported in the current study data generated from a very big cohort, including 675 Presenilin-1 (PSEN1) E280A mutation carriers (141 APOE4+ and 102 APOE2+) and 594 non-carriers (148 APOE4+ and 73 APOE2+). With such big sample size, the authors demonstrated clearly that age-related cognitive decline was accelerated in ADAD mutation carriers who were also APOE4+ compared to those who were APOE4- and delayed in those who were APOE2+ compared to APOE2-. Furthermore, they also showed that education was protective and moderated the effect of APOE genotype on cognition.

This is a timely important study with high impact in APOE and AD research field. Overall, the study was well designed and conducted, the analyses were well powered, and the data support the main conclusions.

This reviewer only has a few minor comments:

1) In Table 1, adding a percentage for each APOE genotype in a parentheses (or a column) should be helpful for readers to more easily follow the data.

2) Although it is under-powered for statistical analyses of the effects of APOE4/4 and APOE2/4 effects, it should still be helpful to show in a supplementary figure the age-related trajectories of cognitive changes for the APOE4/4 and APOE2/4 PSEN1 mutation carriers and non-carriers. It might be very difficult to find a cohort with a even bigger sample size, therefore, it would be interesting for readers to see what the trends look like.

Yadong Huang

We thank the reviewers for their helpful comments and feedback, which we believe have substantially improved our paper. Changes in response to reviewer comments are marked in red in the text, and changes to meet journal formatting requirements are marked in blue. Our point-by-point responses to the reviewers' comments are indicated below in red.

Reviewer #1:

It IS interesting to know the effects of apoe on age at onset and rate of decline in PSEN mutation carriers and this has been addressed before with conflicting results (see also Van Broeckhoven C, Backhovens H, Cruts M, Martin JJ, Crook R, Houlden H, Hardy J. APOE genotype does not modulate age of onset in families with chromosome 14 encoded Alzheimer's disease. *Neurosci Lett.* 1994 Mar 14;169(1-2):179-80. doi: 10.1016/0304-3940(94)90385-9. PMID: 8047278.). This is the largest study to date.

I am not sure however, the authors have correctly interpreted their data. Does apoe genotype influence age at onset (which I would suspect) or rate of decline (which would surprise me). Most studies in non mutation carriers suggest that apoe affects a.o.o. but not rate of decline: ie.disease initiation.

I would suggest reanalysis to distinguish these possibilities

Response: We thank the reviewer for this comment and have revised the manuscript to more accurately and clearly present our results in the following ways:

A subset of *PSEN1* carriers in this dataset received a diagnosis of mild cognitive impairment (MCI) and/or dementia. We examined the age of onset for those participants with MCI and dementia, finding that APOE e4+ genotype is associated with earlier age of clinical onset in *PSEN1* carriers ($ps < .034$). APOE e2 genotype was not statistically significantly related to age of clinical onset, but numerically showed later age of onset for APOE e2+ *PSEN1* carriers ($ps > .339$). These results complement our findings from the age-related cognitive trajectories and suggest that APOE genotype does influence age of onset in *PSEN1* carriers. These results are presented in the Results section (pgs. 6-7) and Supplementary Table 1.

Because we are analyzing only cross-sectional data, we cannot determine whether APOE genotype influences the rate of cognitive decline. We expanded upon this point in the Discussion (pg. 11, "A primary limitation of this study is the reliance of single time-point data. Although our participants spanned a broad range of ages, and age is highly linked to clinical progression in this kindred, we cannot speak to individual trajectories of decline with these data. Analysis of longitudinal cognitive decline will further clarify whether APOE influences age of onset, rate of decline, or both.").

To further clarify our interpretation of the results, we have changed our wording to "accelerated onset of cognitive decline" instead of "accelerated cognitive decline" in the introduction and discussion (e.g., pgs. 4, 8).

Additionally, we have added the Van Broeckhoven et al., 1994 citation to our manuscript.

Reviewer #2:

The study investigated the effect of APOE e4 and e2 status as well as years of education on onset of dementia and cognitive function (MMSE) in a huge cohort of mutation carriers (MC) and non-carriers (NC) from the Colombian PSEN1 E280A kindred. In MC, the results showed a negative e4 and a positive e2 effect and a positive influence of education and the effects were age-related after the onset. In contrast, no significant effects were found in NC.

Given the cross-sectional design, results clarify previous inconsistencies regarding the e4 effect in the Colombian kindred.

In spite of the robustness of the study, some questions about the cohort have come to mind. What was the age range and what was the age distribution? Could the total cohort be divided in two to make it possible to study cohort effects? Could it be that age changes are confounded by cohort changes regarding education, way of living etc? This point was raised by the authors in the discussion.

Response: Thank you to the reviewer for these comments. The ages range from 18 to 75 years for *PSEN1* non-carriers, and from 18 to 73 years for the *PSEN1* carriers. The distribution is shown below:

To address potential confounds with age, we first examined the correlation between age and education in *PSEN1* carriers. Younger age was moderately associated with higher years of education ($r = -0.33$, $p < .001$). We then divided the sample into “young” (18-30 years) and “old” (31+ years) and examined the proportion of rural versus urban living. Young and old *PSEN1* carriers did not differ in type of living. These results are included now in the Supplementary Materials.

A related issue regards “ruralness” that was also mentioned in the discussion as possible important factor to understand differential effects of e4 (positive vs negative). It would be informative to know about urban vs rural living conditions in the cohort. This factor may also be confounded by degree of education.

We identified a subset of participants who had available data on urban versus rural living ($n = 601$). In sum, *PSEN1* carriers who lived in urban areas had higher education than those who lived in rural areas (urban average: 7.70 years, rural average: 6.90 years; $t = 2.04$, $p = .042$). Both APOE e4+ and APOE e4- *PSEN1* carriers had higher proportions of urban versus rural-

living participants, but the proportion of rural-living participants was marginally higher in APOE e4+ *PSEN1* carriers ($p = .051$). Next, we conducted a linear regression with type of living, APOE e4 genotype, and their interaction to predict MMSE scores in *PSEN1* carriers. APOE e4 genotype remained a statistically significant predictor of MMSE score ($p = .043$), but type of living ($p = .080$) and the interaction term between type of living and APOE genotype ($p = .069$) did not reach significance. We reference these analyses in the Discussion (pg. 10) and present them in full in the Supplementary Materials.

In regard to a possible differential effect of e4 on young and old individuals, the concept of "antagonistic pleiotropy" was referred to without presenting a reference. It maybe important for understanding the variability of APOE effects.

We thank the reviewer for pointing this out and have included two references for the concept of antagonistic pleiotropy introduced in the discussion (pg. 9).

Several times (introduction, discussion), the authors use the word "environmental", what is meant by this umbrella term? Pollution, virus, social conditions (SES)? It seems obvious that other factors than AD and APOE genetics and education are players as unknown influence is operating, see Figures 1A, 2A, 3A and 3B (all the dots below regression lines).

To clarify this terminology and more clearly state that multiple factors in addition to education are likely involved, we have made the following edits in our introduction and discussion sections:

"In addition, environmental factors (such as lifestyle, health, and socioeconomic conditions) may influence age-related cognitive trajectories and mitigate genetic risk⁹⁻¹¹. One such factor is education, which has been identified as an important modifiable factor for dementia delay and prevention¹²." (Introduction, pg. 3)

"Finally, education is not the sole environmental factor influencing cognition and clinical progression. More work is needed to further understand the impact of other lifestyle and modifiable factors along with their interactions with genetic makeup." (Discussion, pg. 11)

Reviewer #3:

APOE4 and APOE2 genotype have been identified as a major genetic risk and protective factor, respectively, for late-onset sporadic Alzheimer's disease (AD). However, their effects on autosomal dominant AD (ADAD) are unclear. Previous studies of APOE genotype effects on ADAD yielded inconsistent data due to the under-power analysis related to small sample sizes.

To address this major issue, the authors reported in the current study data generated from a very big cohort, including 675 Presenilin-1 (*PSEN1*) E280A mutation carriers (141 APOE4+ and 102 APOE2+) and 594 non-carriers (148 APOE4+ and 73 APOE2+). With such big sample size, the authors demonstrated clearly that age-related cognitive decline was accelerated in ADAD mutation carriers who were also APOE4+ compared to those who were APOE4- and delayed in those who were APOE2+ compared to APOE2-. Furthermore, they also showed that education was protective and moderated the effect of APOE genotype on cognition.

This is a timely important study with high impact in APOE and AD research field. Overall, the study was well designed and conducted, the analyses were well powered, and the data support the main conclusions.

This reviewer only has a few minor comments:

We thank the reviewer for their kind comments and helpful suggestions to improve this paper.

1) In Table 1, adding a percentage for each APOE genotype in a parentheses (or a column) should be helpful for readers to more easily follow the data.

We have added the percentage for each APOE genotype to Table 1 in parentheses as suggested.

2) Although it is under-powered for statistical analyses of the effects of APOE4/4 and APOE2/4 effects, it should still be helpful to show in a supplementary figure the age-related trajectories of cognitive changes for the APOE4/4 and APOE2/4 PSEN1 mutation carriers and non-carriers. It might be very difficult to find a cohort with a even bigger sample size, therefore, it would be interesting for readers to see what the trends look like.

We included a figure in the Supplementary Materials (Supplementary Figure 2) visualizing the age-related cognitive trajectories of APOE e2/2, e2/4, and e4/4 PSEN1 mutation carriers, which is also referenced in the discussion (pg. 11).

REVIEWERS' COMMENTS

Reviewer #2 (Remarks to the Author):

Based on a large cohort of mutation carriers and non-carriers from the Colombian PSEN1 kindred, the effect of education, APOE e4 +/- and e2 +/- on age and global cognition (MMSE) was investigated. As expected and in line with most previous research, APOE e4+ was a negative factor, e2+ was a positive factor as well as degree of education. This is important to conclude. The study is well done, methods are sound and the study is well written. The study deals with the Colombian PSEN1. In the future, similar studies are needed with other PSEN1 mutation and also mutation in the APP gene. Future studies are also needed regarding the possible antagonistic pleiotropy effect in APOE e4. Three minor points are suggested by the reviewer to be commented upon by the authors. First, it is stated that the participants were blind to their genetic status, but what about the investigators and particularly the investigators who performed the MMSE assessment? Second, why was the possible effect of "ruralness" not investigated? This factor have been suggested in previous research to be linked to the antagonistic pleiotropy. Third, please comment on the possible interaction between APOE e4 and age.

Reviewer #3 (Remarks to the Author):

The authors have addressed all my concerns. This reviewer has no further comment.

Yadong Huang

We thank the reviewers for their time reviewing our manuscript resubmission. Our point-by-point responses to the additional comments from Reviewer 2 are indicated below in red.

Reviewer #2:

Based on a large cohort of mutation carriers and non-carriers from the Colombian PSEN1 kindred, the effect of education, APOE e4 +/- and e2 +/- on age and global cognition (MMSE) was investigated. As expected and in line with most previous research, APOE e4+ was a negative factor, e2+ was a positive factor as well as degree of education. This is important to conclude. The study is well done, methods are sound and the study is well written. The study deals with the Colombian PSEN1. In the future, similar studies are needed with other PSEN1 mutation and also mutation in the APP gene. Future studies are also needed regarding the possible antagonistic pleiotropy effect in APOE e4. Three minor points are suggested by the reviewer to be commented upon by the authors.

First, it is stated that the participants were blind to their genetic status, but what about the investigators and particularly the investigators who performed the MMSE assessment?

Response: The investigators were blind to participant's genetic status throughout data collection, including those who performed the MMSE assessment. We have added this information in the methods section.

Second, why was the possible effect of "ruralness" not investigated? This factor have been suggested in previous research to be linked to the antagonistic pleiotropy.

Response: We were primarily interested in the roles of *APOE* genotype and educational attainment when designing the study. Therefore, we did not originally have a prior hypothesis about the possible effect of ruralness and did not originally examine its effect in the study. Following the reviewer's helpful suggestion to consider this factor in the previous review, we examined the possible effect of ruralness and presented those results in the Supplementary Analysis, as well as in the Discussion. It would be an interesting idea for a future study to more thoroughly examine type of living.

Third, please comment on the possible interaction between APOE e4 and age.

Response: In our Monte Carlo Markov chain models, we found that the effects of *APOE* e4 emerge around age 44. Thus, the effects of *APOE* e4 are dependent on age, such that cognitive scores in younger *PSEN1* carriers do not differ based on *APOE* genotype, but cognitive scores in older *PSEN1* carriers do differ based on *APOE* genotype. Importantly, the non-carrier control groups did not exhibit any effects of *APOE* genotype and were included as an age-matched control group. This suggests that our effects are related more so to disease progression than to age itself.